# Real world external validation of metabolic gestational age assessment in Kenya

Steven Hawken[1,2], Victoria Ward[3], A. Brianne Bota[1], Monica Lamoureux[4], Robin Ducharme[1], Lindsay A. Wilson[1], Nancy Otieno[5], Stephen Munga[5], Bryan O. Nyawanda[5], Raphael Atito[5], David K. Stevenson[3], Pranesh Chakraborty[4,6], Gary L. Darmstadt[3‡], Kumanan Wilson[1,7,8‡]*

1 Clinical Epidemiology Program, Ottawa Hospital Research Institute, Ottawa, Canada, 2 School of Epidemiology and Public Health, University of Ottawa, Ottawa, Canada, 3 Prematurity Research Center, Department of Pediatrics, Stanford University School of Medicine, Stanford, California, United States of America, 4 Newborn Screening Ontario, Children's Hospital of Eastern Ontario, Ottawa, Canada, 5 Kenya Medical Research Institute (KEMRI), Center for Global Health Research, Kisumu, Kenya, 6 Departments of Pediatrics, and of Biochemistry, Microbiology and Immunology, University of Ottawa, Ottawa, Canada, 7 Department of Medicine, University of Ottawa, Ottawa, Canada, 8 Bruyère Research Institute, Ottawa, Canada

‡ GLD and KW contributed equally to this project as co-senior authors.
* kwilson@ohri.ca

**Data Availability Statement:** De-identified data related to this manuscript are publicly available

## Abstract

Using data from Ontario Canada, we previously developed machine learning-based algorithms incorporating newborn screening metabolites to estimate gestational age (GA). The objective of this study was to evaluate the use of these algorithms in a population of infants born in Siaya county, Kenya. Cord and heel prick samples were collected from newborns in Kenya and metabolic analysis was carried out by Newborn Screening Ontario in Ottawa, Canada. Postnatal GA estimation models were developed with data from Ontario with multivariable linear regression using ELASTIC NET regularization. Model performance was evaluated by applying the models to the data collected from Kenya and comparing model-derived estimates of GA to reference estimates from early pregnancy ultrasound. Heel prick samples were collected from 1,039 newborns from Kenya. Of these, 8.9% were born preterm and 8.5% were small for GA. Cord blood samples were also collected from 1,012 newborns. In data from heel prick samples, our best-performing model estimated GA within 9.5 days overall of reference GA [mean absolute error (MAE) 1.35 (95% CI 1.27, 1.43)]. In preterm infants and those small for GA, MAE was 2.62 (2.28, 2.99) and 1.81 (1.57, 2.07) weeks, respectively. In data from cord blood, model accuracy slightly decreased overall (MAE 1.44 (95% CI 1.36, 1.53)). Accuracy was not impacted by maternal HIV status and improved when the dating ultrasound occurred between 9 and 13 weeks of gestation, in both heel prick and cord blood data (overall MAE 1.04 (95% CI 0.87, 1.22) and 1.08 (95% CI 0.90, 1.27), respectively). The accuracy of metabolic model based GA estimates in the Kenya cohort was lower compared to our previously published validation studies, however inconsistency in the timing of reference dating ultrasounds appears to have been a contributing factor to diminished model performance.

through Dryad: https://doi.org/10.5061/dryad.
wwpzgmsmv.

**Funding:** This research was supported by grants
from the Bill & Melinda Gates Foundation awarded
to KW (award number: OPP1184574) and GLD
(award number: OPP1182996). The funders had
no role in study design, data collection and
analysis, decision to publish, or preparation of the
manuscript.

**Competing interests:** The authors have declared
that no competing interests exist.

## Introduction

The need for novel, non-invasive methods to accurately estimate gestational age (GA) in low resource settings has been identified by the World Health Organization as a priority area for improving global estimation of the burden of preterm birth at < 37 completed weeks of gestation. Preterm birth as well as being born small (small for gestational age; SGA = lowest ten centiles of birthweight given gestational age) are leading causes of infant mortality and morbidity, particularly in low- and middle-income countries (LMIC) [1, 2]. Furthermore, medical needs and developmental milestones differ between term, preterm and SGA infants. Thus, accurately identifying at-risk infants at birth is important, both for estimating the true population burden of preterm birth, and potentially in informing postnatal care and supportive resources. Although the use of first-trimester ultrasound has improved our ability to estimate GA [3], it is not widely available in all low resource settings and its implementation poses significant obstacles, including cost, training, equipment maintenance and lack of standardization [4]. In low resource settings without access to prenatal ultrasound, GA estimates are often made based on last menstrual period, the accuracy of which may be affected by memory recall as well as irregular menses and maternal malnutrition [5, 6]. Commonly used postnatal examination methods for GA dating of infants (e.g., Dubowitz or Ballard score) also have limitations in terms of their accuracy–particularly in preterm and growth-restricted infants–and their utility is further limited by challenges with feasibility and high inter-user variability [7].

Given the limitations associated with existing GA dating methods, numerous research groups are testing new ways to accurately estimate GA [8–11]. We have developed novel machine learning-based algorithms that use newborn screening metabolites and clinical and demographic covariates to estimate GA [12, 13]. These algorithms were originally developed and internally validated in a large cohort of newborns in Ontario, Canada [14, 15]. Refinements to the algorithms incorporated machine learning and improved the accuracy of gestational age estimations [12, 16]. Here we evaluate the use of these algorithms in a population of infants born in Siaya County, Kenya.

## Methods

### Ethics

This study was approved by the Ottawa Health Sciences Network Research Ethics Board (20180330-01H), Children's Hospital of Eastern Ontario Research Ethics Board (18/58X), the Stanford University School of Medicine Institutional Review Board (44656) and the Kenya Medical Research Institute (KEMRI) Scientific and Ethics Review Unit (SSC 2880).

### Study setting

A detailed study protocol has previously been published which describes the study sites and provides further details on sample collection and processing [17]. The Kenya study site is located in Kisumu at the KEMRI Centre for Global Health Research, with field sites located in Siaya County, where a maternal-infant demographic surveillance program followed a prospective cohort of pregnant women and their infants in two community hospitals: Siaya County Referral Hospital (SCRH) and Bondo sub-County Hospital (BSCH). Eligible participants were pregnant women between the ages of 15–49 years, residing within a 10 km radius of the research facility, willing to deliver in the research hospital, and not planning to relocate within 1 year of enrollment into the surveillance program. Participants were enrolled at their first antenatal care visit (ANC-1), which typically occurred prior to 20 weeks' gestation. Participants underwent an early pregnancy ultrasound as early as possible and were offered treatment

for common illnesses, including malaria, urinary tract infections, and sexually transmitted infections. A small portion of infants were born at home and evaluated within 72 hours of delivery.

## Consent

Informed written consent was obtained from all mothers prior to study enrollment. All live-born infants of enrolled mothers were eligible for inclusion.

## Collection of newborn screening specimens

Cord blood samples were collected via syringe within 30 minutes of delivery of the placenta. Four to five drops of blood from the syringe were applied to filter paper within pre-printed circles. Heel prick samples were collected from newborns ideally between 24–72 hours after birth, or prior to discharge if the newborn was released from the hospital within 24 hours of delivery. The newborn's heel was warmed prior to skin puncture to promote blood flow. The puncture site was cleaned and air-dried and a sterile lancet was used to puncture the lateral plantar aspect of the newborn's heel. The first drop of blood was wiped away and 4–5 drops of blood were applied within pre-printed circles of a second filter paper.

Heel and cord dried blood spot (DBS) cards were dried and stored at ambient temperature and shipped weekly to the Newborn Screening Ontario (NSO) laboratory at the Children's Hospital of Eastern Ontario in Ottawa, Canada for analysis, along with clinical and demographic information required for clinical interpretation of metabolic profiles and for metabolic GA estimation models. This information included infant sex, birthweight (in grams), multiple birth status, GA (in weeks + days), date of birth, and timing of sample collection.

## Newborn screening analysis

The newborn screening analysis process has been described in detail previously [17]. Dried blood spot samples were analyzed for the following metabolites: hemoglobin profiles, 17-hydroxyprogesterone (17-OHP), thyroid stimulating hormone (TSH), immunoreactive trypsinogen (IRT), a panel of 12 amino acids and 31 acylcarnitines, T-cell receptor excision circles (TREC), biotinidase activity, and galactose-1-phosphate uridylyltransferase activity (Table 1). Real-time screening for three conditions [congenital hypothyroidism (CH), hemoglobinopathies, and medium-chain acyl-CoA dehydrogenase deficiency (MCADD)] occurred during this study. These conditions were deemed to be high priority for reporting and were treatable at the local collection sites. Results of screening for congenital metabolic conditions will be published elsewhere.

**Table 1. Newborn screening analytes included in predictive models.**

| Hemoglobins | Adult hemoglobin: HbA(A) |
|---|---|
| | Fetal hemoglobin: HbF (F), Acetylated HbF (F1) |
| Endocrine markers | 17-hydroxyprogesterone (17-OHP), Thyroid stimulating hormone (TSH) |
| Amino Acids | Arginine (arg); phenylalanine (phe); alanine (ala); leucine (leu); ornithine (orn); citruline (cit); tyrosine (tyr); glycine (gly); methionine (met); valine (val); |
| Acyl-carnitines | C0; C2; C3; C4; C5; C5:1; C6; C8; C8:1; C10; C10:1; C12; C12:1; C14; C14:1; C14:2; C16; C18; C18:1; C18:2; C10:1; C12:1; C14:1; C14:2; C4OH; C5:1; C5DC; C5OH; C6DC; C16:OH; C16:1OH; C18OH; C18:1OH; C3DC; C4DC |
| Enzyme markers | Biotinidase; immunotripsinogen; galactose-1-phosphate uridylyltransferase (GALT) |
| T-cell Function | T-cell Receptor Excision Circles (TREC) |

### Data preparation and statistical analysis

All analyses were conducted using SAS 9.4 [18] and R 3.3.2 [19]. Data preparation steps, including standardization and log transformations are detailed in S1 Appendix.

Analytes were included as candidate predictors in GA estimation models based on their routine measurement as part of Ontario's expanded newborn screening program, including hemoglobin profiles, amino acids, acylcarnitines, hormone and endocrine markers, enzymes and co-enzymes (Table 1). Newborn GA was estimated from models derived using multivariable regression coupled with elastic net regularization and including the following covariates:

1. Model 1: Birth weight, sex, multiple birth status and pairwise interactions.

2. Model 2: Birth weight, sex, multiple birth status and newborn screening analytes and pairwise interactions.

Models were trained and internally validated in independent training and validation/test cohorts of infants from Ontario, Canada (S1 Appendix). These pretrained models were then applied to the data for infants from the external cohorts to estimate GA. To evaluate model accuracy, GA estimates were compared to the ultrasound reference GA for each infant, and the residual error calculated. Different metrics were calculated to estimate model uncertainty, including mean square error (MSE), standard error of estimation [also known as root mean square error (RMSE)], and mean absolute error (MAE), which is the average of the absolute value of the residual across all subjects (or subsets of subjects). MAE is less sensitive to outliers (large residuals) and is a helpful metric of "average error" which is often reported in model validation studies with continuous outcomes. We have included both metrics to facilitate comparisons with other study results. Additionally, we calculated the proportion of model-derived estimates that fell within ± 1 week of reference GA. MAE was the main performance metric used to evaluate model accuracy, but multiple metrics were calculated and reported to facilitate comparisons to other models developed by our group and others. Model-derived frequency of preterm birth will be compared to the observed prevalence of preterm birth.

## Results

1,039 newborns had heel prick samples available, as well as clinical and demographic data including ultrasound-derived reference GA. Of these, 92 infants (8.9%) were preterm and 88 (8.5%) were SGA (Table 2). 1,012 newborns (97.4%) also had a cord blood sample collected. It should be noted that the Ontario cohort in which the models were developed and internally validated had a lower preterm birth prevalence of 5.6% and SGA prevalence of 3.9% (Table 2). There were 11 screen positive results for hemoglobinopathies that were reported back to the study site for follow up. Details of incidental screen positive findings and follow up will be reported in a separate manuscript.

### Model-based GA estimation for heel prick samples

Overall, Model 1, which included only readily available clinical covariates (sex, birthweight and multiple birth status) estimated GA within 10.5 days on average, with a MAE of 1.5 (95% CI 1.41, 1.58) weeks. 58.5% of model estimates were within ± 1 week of reference GA. For preterm births, Model 1 MAE was 2.64 (95% CI 2.30,3.01) weeks and only 24.1% of estimates were within ± 1 week of reference GA. In SGA newborns, MAE was 3.13 (95% CI 2.85, 3.38) weeks and 3.4% of estimates were within ± 1 week of reference GA (Table 2). Model 2, which included clinical covariates plus analytes, estimated GA within 9.5 days overall, with a MAE of 1.35 (95% CI 1.27, 1.43) weeks, and 64.1% of estimates were within ± 1 week of reference GA.

**Table 2. Cohort characteristics.**

| | Canada n = 39, 666 (Ontario test cohort) | Kenya Heel Prick n = 1,039 | Kenya Cord Blood n = 1,012 |
|---|---|---|---|
| **Sex, n(%)** | | | |
| Male | 19,536 (49.3%) | 526 (50.6%) | 511 (50.5%) |
| Female | 20,130 (50.5%) | 513 (49.4%) | 501 (49.5%) |
| **Birth weight (g), mean ± SD** | | | |
| Overall | 3,379 ± 530.2 | 3,238.4 ± 468.9 | 3,238.5 ±470.1 |
| Term infants only | 3,430.6 ± 476.1 | 3,277.1 ± 430.6 | 3,274.6 ± 435.1 |
| Preterm infants only | 2,504.1 ± 622.8 | 2,840.0 ± 635.4 | 2869.0 ± 632.5 |
| **Low birth weight (<2500 g), n (%)** | 1,812 (4.6%) | 48 (4.6%) | 46 (4.6%) |
| **LGA (>90th Centile), n (%)** | 3983 (10.0%) | 139 (13.4%) | 139 (13.7%) |
| **SGA (<10th Centile), n (%)** | 1,561 (3.94%) | 88 (8.5%) | 87 (8.6%) |
| **SGA (<3rd Centile), n (%)** | 363 (0.92%) | 28 (2.7%) | 30 (3.0%) |
| **Completed gestational age wks, mean ± SD** | 39.3±1.6 | 39.1 ±1.9 | 39.1±1.9 |
| Term (≥37 wks), n (%) | 37,440 (94.4%) | 947 (91.1%) | 922 (91.1%) |
| Late Preterm (32–36 wks), n (%) | 2,049 (5.2%) | 88 (8.5%) | 86 (8.5%) |
| Very Preterm (28–31 wks), n (%) | 126 (0.3%) | 4 (0.4%) | 4 (0.4%) |
| Extremely Preterm (<28 wks), n (%) | 51 (0.1%) | 0 | 0 |

SGA, small for gestational age (lowest 10 and 3 centiles); LGA, large for gestational age (highest 10 Centiles) within gestational age and sex strata, calculated using Intergrowth-21 centiles and applied uniformly in the Ontario and Kenya cohorts.

In preterm infants, MAE was 2.62 (95% CI 2.28, 2.99) weeks and in SGA infants the MAE was 1.81 (95% CI 1.57, 2.07) weeks (Table 3).

The performance of Models 1 and 2 did not appear to be affected by the HIV status of the mother. Results for subjects with HIV-positive mothers (n = 197) were almost identical to model performance for infants of HIV-negative mothers (n = 842) (Table 4).

## Model-based GA estimation for cord-blood samples

Model 1 demonstrated nearly identical performance in cord blood samples compared to heel prick samples, as analytes were not covariates in Model 1, and the heel and cord blood cohorts were almost entirely comprised of the same infants. Overall, in the cord blood cohort, Model 2

**Table 3. Summary of model performance to estimate gestational age in samples from Kenya.**

| Models | Kenya Heel Prick Samples | | | Kenya Cord Blood Samples | | |
|---|---|---|---|---|---|---|
| | Overall, N = 1,039 | Preterm, N = 93 | SGA, N = 88 | Overall, N = 1,012 | Preterm, N = 91 | SGA, N = 87 |
| *Model 1: Sex and birth weight* | | | | | | |
| MAE (CI) | 1.50 (1.41, 1.58) | 2.64 (2.30, 3.01) | 3.13 (2.85, 3.38) | 1.51 (1.42, 1.59) | 2.70 (2.39, 3.04) | 3.17 (2.91, 3.46) |
| RMSE (CI) | 2.02 (1.91, 2.13) | 3.09 (2.75, 3.45) | 3.39 (3.06, 3.70) | 2.04 (1.94, 2.16) | 3.14 (2.84, 3.49) | 3.43 (3.10, 3.77) |
| % +/-1 wk (CI) | 58.5 (55.3, 61.6) | 24.1 (14.5, 34.1) | 3.4 (0.0, 7.7) | 58.4 (55.4, 61.6) | 23.4 (13.7, 32.0) | 3.6 (0.0, 8.3) |
| *Model 2: Sex, Birth weight and analytes* | | | | | | |
| MAE (CI) | 1.35 (1.27, 1.43) | 2.62 (2.28, 2.99) | 1.81 (1.57, 2.07) | 1.44 (1.36, 1.53) | 2.79 (2.46, 3.12) | 2.06 (1.76, 2.36) |
| RMSE (CI) | 1.83 (1.72, 1.94) | 3.09 (2.74, 3.48) | 2.18 (1.91, 2.46) | 1.95 (1.85, 2.06) | 3.19 (2.85, 3.57) | 2.41 (2.13, 2.69) |
| % +/-1 wk (CI) | 64.1 (61.1, 67.2) | 29.4 (20.2, 38.9) | 46.9 (35.8, 57.3) | 61.2 (58.3, 63.9) | 21.1 (11.8, 29.3) | 33.2 (23.0, 44.6) |

MAE: Mean absolute error; RMSE: Root mean square error; SGA: small for gestational age

Data are presented as the mean and 2.5th and 97.5th bootstrap percentiles for MAE, RMSE and the percentage of model estimates within 1 and 2 weeks of ultrasound GA for 1000 bootstrap samples generated from each cohort.

**Table 4. Summary of model performance to estimate gestational age in heel prick samples according to HIV status and timing of GA dating ultrasound.**

| | Kenya Heel Prick Samples | | | | | |
|---|---|---|---|---|---|---|
| | HIV Neg, N = 842 | HIV Pos, N = 197 | US <9 weeks, N = 28 | US 9–13 weeks, N = 120 | US 14–20 weeks, N = 503 | US >20 weeks, N = 386 |
| **Model 1: Sex, birthweight multiple birth** | | | | | | |
| MAE (CI) | 1.49 (1.39, 1.57) | 1.56 (1.37 1.74) | 1.51 (1.11, 1.93) | 1.27 (1.08, 1.45) | 1.57 (1.45, 1.69) | 1.48 (1.33, 1.64) |
| RMSE (CI) | 2.01 (1.89, 2.13) | 2.06 (1.81, 2.31) | 1.89 (1.53, 2.29) | 1.61 (1.40, 1.83) | 2.07 (1.92, 2.23) | 2.08 (1.89, 2.28) |
| % +/-1 wk (CI) | 58.1 (54.8, 61.7) | 60.4 (53.9, 67.0) | 49.6 (32.1, 67.7) | 68.6 (60.2, 76.3) | 55.5 (51.0, 60.0) | 60.0 (54.6, 65.0) |
| **Model 2: Sex, birthweight multiple birth and analytes** | | | | | | |
| MAE (CI) | 1.34 (1.27, 1.43) | 1.35 (1.16, 1.53) | 1.48 (1.09, 1.90) | 1.04 (0.87, 1.22) | 1.34 (1.23, 1.46) | 1.43 (1.31, 1.58) |
| RMSE (CI) | 1.81 (1.70, 1.93) | 1.93 (1.67, 2.20) | 1.85 (1.46, 2.23) | 1.38 (1.19, 1.58) | 1.84 (1.68, 2.00) | 1.94 (1.77, 2.13) |
| % +/-1 wk (CI) | 63.7 (60.5, 66.9) | 65.7 (59.5, 72.0) | 49.6 (29.6, 66.7) | 71.5 (63.3, 79.1) | 64.5 (59.8, 68.9) | 62.2 (56.8, 66.8) |

MAE: Mean absolute error; RMSE: Root mean square error; SGA: small for gestational age

Data are presented as the mean and 2.5th and 97.5th bootstrap percentiles for MAE, RMSE and the percentage of model estimates within 1 and 2 weeks of ultrasound GA for 1000 bootstrap samples generated from each cohort.

had a MAE of 1.44 (95% CI 1.36, 1.53) weeks. In preterm infants, Model 2 had a MAE of 2.79 (95% CI 2.46, 3.12) weeks and in SGA infants the MAE was 2.06 (95% CI 1.76, 2.36) weeks (Table 2). Like the heel prick results, model performance was not sensitive to the HIV status of the mother (Tables 5 and 6).

## Model-based GA estimation using reference GA derived from ultrasounds within recommended window

There was significant variation in the timing of gestational dating ultrasound, despite best efforts to conduct the ultrasound as early as possible. Reference GA for 28 newborns (2.7%) was derived from ultrasound conducted before 9 weeks' gestation, and 889 (85.6%) had reference GA based on an ultrasound later than 13 weeks' gestation. Only 120 newborns (11.5%) had reference GA based on an ultrasound conducted within 9–13 weeks' gestation (Table 4). When evaluated in these 120 newborns, model performance was markedly better, with Model 2 having a MAE of 1.04 (95% CI 0.87, 1.22) weeks overall and a MAE of 2.56 (95% CI 1.50, 4.00) and 1.07 (95% CI 0.46, 1.70) weeks in preterm and SGA infants, respectively (Table 5).

**Table 5. Summary of model performance in cord blood samples according to HIV status and timing of GA dating ultrasound (US).**

| Models | HIV Neg, N = 822 | HIV Pos, N = 190 | US <9 weeks, N = 27 | US 9–13 weeks, N = 121 | US 14–20 weeks, N = 494 | US >20 weeks, N = 370 |
|---|---|---|---|---|---|---|
| **Model 1: Sex, birthweight and multiple birth** | | | | | | |
| MAE (CI) | 1.49 (1.41, 1.59) | 1.56 (1.37, 1.76) | 1.41 (0.96, 1.89) | 1.28 (1.10, 1.45) | 1.58 (1.46, 1.71) | 1.48 (1.33, 1.63) |
| RMSE (CI) | 2.03 (1.92, 2.15) | 2.08 (1.86, 2.34) | 1.84 (1.43, 2.26) | 1.63 (1.39, 1.84) | 2.10 (1.94, 2.25) | 2.09 (1.88, 2.27) |
| % +/-1 wk (CI) | 57.9 (54.3, 60.9) | 60.5 (53.3, 67.0) | 55.6 (35.7, 73.9) | 67.5 (59.2, 76.0) | 55.0 (50.8, 59.4) | 60.1 (54.7, 64.6) |
| **Model 2: Sex, birthweight, multiple birth, and analytes** | | | | | | |
| MAE (CI) | 1.43 (1.34, 1.52) | 1.49 (1.29, 1.70) | 1.60 (1.22, 2.03) | 1.08 (0.90, 1.27) | 1.46 (1.33, 1.57) | 1.53 (1.39, 1.66) |
| RMSE (CI) | 1.92 (1.81, 2.04) | 2.04 (1.80, 2.28) | 1.89 (1.49, 2.30) | 1.44 (1.25, 1.62) | 1.99 (1.83, 2.14) | 2.03 (1.87, 2.19) |
| % +/-1 wk (CI) | 61.4 (58.3, 64.4) | 60.4 (53.1, 67.6) | 55.4 (36.7, 73.1) | 70.5 (62.2, 78.2) | 61.7 (57.5, 65.7) | 57.9 (53.3, 62.6) |

MAE: Mean absolute error; RMSE: Root mean square error; SGA: small for gestational age

Data are presented as the mean and 2.5th and 97.5th bootstrap percentiles for MAE, RMSE and the percentage of model estimates within 1 and 2 weeks of ultrasound GA for 1000 bootstrap samples generated from each cohort.

**Table 6. Summary of model performance in heel and cord blood samples restricted to 9–13 week ultrasound.**

| Models | Kenya Heel Prick Samples | | | Kenya Cord Blood Samples | | |
|---|---|---|---|---|---|---|
| | Overall, N = 120 | Preterm, N = 5 | SGA, N = 13 | Overall, N = 120 | Preterm, N = 5 | SGA, N = 14 |
| **Model 1: Sex, birthweight and multiple birth** | | | | | | |
| *MAE (CI)* | 1.27 (1.08, 1.45) | 2.80 (1.00, 4.50) | 2.47 (1.92, 3.00) | 1.27 (1.08, 1.47) | 2.82 (1.20, 4.33) | 2.52 (2.08, 3.00) |
| *RMSE (CI)* | 1.61 (1.40, 1.83) | 3.19 (1.41, 4.53) | 2.63 (2.02, 3.22) | 1.62 (1.39, 1.87) | 3.22 (1.73, 4.43) | 2.67 (2.17, 3.22) |
| *% +/-1 wk (CI)* | 68.6 (60.2, 76.3) | 18.8 (0.0, 60.0) | 7.8 (0.0, 27.3) | 67.6 (59.2, 76.7) | 19.6 (0.0, 66.7) | 7.5 (0.0, 25.0) |
| **Model 2: Sex, birthweight, multiple birth and analytes** | | | | | | |
| *MAE (CI)* | 1.04 (0.87, 1.22) | 2.56 (1.50, 4.00) | 1.07 (0.46, 1.70) | 1.07 (0.88, 1.23) | 2.60 (1.00, 4.00) | 1.52 (0.88, 2.15) |
| *RMSE (CI)* | 1.38 (1.19, 1.58) | 2.78 (1.58, 4.00) | 1.44 (0.73, 2.02) | 1.43 (1.24, 1.62) | 2.94 (1.63, 4.00) | 1.90 (1.30, 2.37) |
| *% +/-1 wk (CI)* | 71.5 (63.3, 79.1) | 18.8 (0.0, 60.0) | 77.0 (50.0, 100.0) | 71.0 (62.5, 79.2) | 19.6 (0.0, 66.7) | 49.4 (22.2, 75.0) |

MAE: Mean absolute error; RMSE: Root mean square error; SGA: small for gestational age

Data are presented as the mean and 2.5th and 97.5th bootstrap percentiles for MAE, RMSE and the percentage of model estimates within 1 and 2 weeks of ultrasound GA for 1000 bootstrap samples generated from each cohort.

Similar to the heel prick results, Model 2 for cord blood specimens performed markedly better when only samples with reference GA ascertained between 9 and 13 weeks of gestation were included [overall MAE 1.08 (95% CI 0.90, 1.27) weeks] (Table 5).

## Discussion

We externally validated the performance of a postnatal GA dating algorithm developed and validated in a cohort of infants in Ontario, Canada in a prospective birth cohort in Siaya County Kenya, a lower-middle-income sub-Saharan African country. Heel prick and umbilical cord blood samples were collected shortly after birth, and ultrasound was used to provide a reference GA for each infant. Overall, model performance was worse in the Kenya birth cohort for Model 1 and Model 2 compared to internally validated model performance in Ontario, and in comparison to previously published external validations of metabolic GA algorithms [12, 13]. The heterogeneity of reference ultrasound timing was an important contributor to diminished model performance, as only 120 out of 1,039 participants had reference ultrasound completed between 9 and 13 weeks of gestation. Model performance was markedly better in participants with reference GA ascertained inside compared to outside the recommended window. For example, Model 2 had an overall MAE of 1.04 weeks among infants with reference GA between 9 and 13 weeks, compared to MAEs of 1.48 (<9 weeks), 1.34 (14–20 weeks) and 1.43 (>20 weeks) weeks for those with dating ultrasounds earlier and later than the recommended window. A similar pattern was seen for Model 1 and Model 2 in heel and cord samples both overall and in preterm and SGA newborns. Our study highlights the challenges in reliably estimating GA in low resource settings, even in those with access to dating ultrasound, given that the timing of dating ultrasound is critical to accurate estimations of GA [3, 20]. Indeed, most pregnant women in Kenya access ANC for the first time in the second trimester [21]. These challenges further underscore the need for novel, reliable GA estimation methods that can be adopted in LMICs.

Given the significant barriers to obtaining an early dating ultrasound, the metabolic GA approach may be a more feasible alternative approach to GA dating than dating ultrasound when the timing of the latter is variable. Our study also demonstrated the utility of cord blood samples, which could further strengthen the feasibility of our approach in low resource settings. Cord blood samples are obtained shortly after birth and remove the burden of sample collection before discharge, do not cause any discomfort to the newborn and may be more

readily accepted by parents who are not accustomed to the heel prick procedure. Given the higher prevalence of HIV in our patient population, our results also provide reassurance that HIV positive status does not appear to impact performance of algorithms based only on clinical measurements (Model 1) or those including metabolic markers measured in heel prick or cord blood (Model 2).

The major limitation of our study was the small number of GA ultrasounds conducted during the optimal reference time-period. Therefore, a gold standard for reliable comparison with accurate true GA was not possible for a large percentage of the sample. Further, as observed in our previous validation studies, the study sample was affected by a participation bias against preterm and extremely preterm infants. Model estimated gestational ages were most accurate in infants born close to full-term, and were overestimated in preterm infants and underestimated in post-term infants. Strengths of the study include the real-world approach to evaluating the algorithm, allowing us to assess not only model performance but the feasibility of this GA estimation approach as well.

Our study demonstrated that, despite being conducted within a prospective pregnancy cohort with a well-defined protocol in a controlled research setting, there were still challenges in obtaining a true reference GA measurement due to timing of dating ultrasounds. Even under perfect circumstances, the metabolic prediction algorithm may not agree perfectly with the ultrasound-based GA because it is based on "metabolic maturity" rather than physical size, which may in fact be a better marker of physiological maturity. The results of this evaluation suggest that postnatal GA estimation algorithms such as the ones we have developed are both feasible and accurate, and previous analyses have indicated that GA estimation algorithm approaches are also potentially cost-effective [22]. Therefore, we believe that GA estimation algorithms based on metabolic analysis of heel prick or cord blood DBS may be able to serve an important role in both individual infant estimates of GA and population level estimations of preterm birth rates. Algorithm-based GA estimates have potential even in settings where early ultrasound is available, given the substantial heterogeneity in timing of reference GA ultrasound in our population, a factor that may compromise the accuracy of estimates based on ultrasound alone. Given these findings, we believe that GA estimation algorithms may serve an important role in providing both individual estimates of GA and population-level estimates of preterm birth.

## Supporting information

**S1 Appendix. Supplementary methods.**
(DOCX)

## Author Contributions

**Conceptualization:** Steven Hawken, Victoria Ward, Nancy Otieno, Pranesh Chakraborty, Gary L. Darmstadt, Kumanan Wilson.

**Data curation:** Steven Hawken, Monica Lamoureux, Robin Ducharme.

**Formal analysis:** Steven Hawken.

**Funding acquisition:** Gary L. Darmstadt, Kumanan Wilson.

**Investigation:** Steven Hawken, Victoria Ward, A. Brianne Bota, Monica Lamoureux, Robin Ducharme, Nancy Otieno, Stephen Munga, Bryan O. Nyawanda, Raphael Atito, Pranesh Chakraborty.

**Methodology:** Steven Hawken, Victoria Ward, Monica Lamoureux, Stephen Munga, Bryan O. Nyawanda, Pranesh Chakraborty, Gary L. Darmstadt, Kumanan Wilson.

**Project administration:** Victoria Ward, A. Brianne Bota, Monica Lamoureux, Robin Ducharme, Lindsay A. Wilson, Nancy Otieno, Bryan O. Nyawanda.

**Resources:** A. Brianne Bota, Monica Lamoureux, Lindsay A. Wilson, Stephen Munga, Raphael Atito, David K. Stevenson.

**Software:** Steven Hawken.

**Supervision:** Victoria Ward, Nancy Otieno, Pranesh Chakraborty, Gary L. Darmstadt, Kumanan Wilson.

**Validation:** Steven Hawken.

**Visualization:** Gary L. Darmstadt, Kumanan Wilson.

**Writing – original draft:** Steven Hawken, A. Brianne Bota.

**Writing – review & editing:** Steven Hawken, Victoria Ward, A. Brianne Bota, Monica Lamoureux, Robin Ducharme, Lindsay A. Wilson, Nancy Otieno, Stephen Munga, Bryan O. Nyawanda, Raphael Atito, David K. Stevenson, Pranesh Chakraborty, Gary L. Darmstadt, Kumanan Wilson.

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
