## [Decision Letter · Decision Letter 0]

28 Jun 2022

PGPH-D-22-00803

Real world external validation of metabolic gestational age assessment in Kenya

Dear Dr. Wilson,

Thank you for submitting your manuscript to PLOS Global Public Health. After careful consideration, we feel that it has merit but does not fully meet PLOS Global Public Health’s publication criteria as it currently stands. Therefore, we invite you to submit a revised version of the manuscript that addresses the points raised during the review process.

The manuscript has received positive responses for both the reviewers. But, they have raised some important doubts, which upon clarification will add value the manuscript.

Please submit your revised manuscript by . If you will need more time than this to complete your revisions, please reply to this message or contact the journal office at globalpubhealth@plos.org. Please include the following items when submitting your revised manuscript:

We look forward to receiving your revised manuscript.

Kind regards,

Ramachandran Thiruvengadam, M.D.,

Academic Editor

Journal Requirements:

1. Please update your online Competing Interests statement. If you have no competing interests to declare, please state: “The authors have declared that no competing interests exist.”

2. We have noticed that you have uploaded Supporting Information files, but you have not included a list of legends. Please add a full list of legends for your Supporting Information files after the references list.

Additional Editor Comments (if provided):

The manuscript has received positive responses for both the reviewers. But, they have raised some important doubts, which upon clarification will add value the manuscript.

Reviewers' comments:

Reviewer's Responses to Questions

**Comments to the Author**

1. Does this manuscript meet PLOS Global Public Health’s publication criteria? Is the manuscript technically sound, and do the data support the conclusions? The manuscript must describe methodologically and ethically rigorous research with conclusions that are appropriately drawn based on the data presented.

Reviewer #1: Yes

Reviewer #2: Yes

2. Has the statistical analysis been performed appropriately and rigorously?

Reviewer #1: Yes

Reviewer #2: Yes

3. Have the authors made all data underlying the findings in their manuscript fully available (please refer to the Data Availability Statement at the start of the manuscript PDF file)?

Reviewer #1: Yes

Reviewer #2: Yes

4. Is the manuscript presented in an intelligible fashion and written in standard English?

Reviewer #1: Yes

Reviewer #2: Yes

5. Review Comments to the Author

Reviewer #1: This manuscript is addressing a fundamental problem in maternal-child global health which is how can the gestational age (GA) at birth of an infant be determined in a low resource setting where a gold standard of first trimester US is impractical. Fundal height, LMP, Ballard like scores all have large inaccuracies, especially in the face of infants being SGA but recently biochemical phenotyping of the mother and/or fetus have shown promise. This group and others have previously shown the data from newborn screening can provide an accurate estimate and this provides a reasonable alternative if the data is needed to determine the population prevalence of preterm birth, the range of gestational ages for newborns or whether a new intervention is changing the gestational age at delivery. In this study the investigators look to see if methods and algorithms developed in high resource settings can be used reliably in an LMIC setting (Kenya in this case). The report adds important data to the struggle to find accurate and effective tools to determine GA in LMICs. Overall the data is well collected and presented and the report well analyzed and written and appropriately cautious in its conclusions. There are a few concerns.

The last sentence of the abstract could be written with more specificity and impact. We have known for at least 3 decades that early US provides the best estimates of GA and any future studies in LMICs will likely use that information in a research setting, as they have done, with a study design that would target early US as the gold standard. Rather the conclusion should be what this study specifically shows.

While the authors rightly point out that “medical needs and developmental milestones differ between term, preterm and SGA infants and thus, accurately identifying at-risk infants at birth is important in informing their postnatal care” the use of metabolic dating done after birth will have limited utility for the medical management of newborns as the results will come in a couple of days at best after testing when the most critical needs will have been addressed by the clinical state of the infant. It would seem that most utility of such testing would be to determine population frequencies of GA or preterm birth rates and/or to determine if improvements in PTB or SGA rates have occurred after interventional studies addressing these needs are underway. They do a better job of this in the discussion. Nonetheless it’s a bit confusing as to whether the conclusion here is that we need a higher percentage of early US or that metabolic screening works. Its not clear why they could not get more compliance around appropriately timed US given that they describe this as a prospective pregnancy cohort. There are many other LMIC studies in which surveillance is done for timing of pregnancy at two or even one month intervals where detection by missed LMP can then lead to US at an optimal time.

I am unable to address issues related to the references cited in that the version of the manuscript I had to review (sent twice) were missing most of the references that should have been on pages 8 and 9. Appropriate citations will need to be assessed prior to acceptance and publication.

In the Bota methods reference paper it refers to plans to analyze based on 3 models but only 2 are reported here. Why the change?

Why do they include multiple births at all in this study and then adjust for them? Its well known the twins and above deliver earlier than singletons and this seems an unnecessary confounder for a study of this type.

They appear to have switched from MRE to MAE as a primary metric in reporting. Is there a reason for that?

In the supplement they refer to 159,000 infants being used in model development from Ontario but in the primary table its 39,000. What is the difference?

Minor Comments:

In the Bota, Gates Open Research paper the PTB rate in Kenya was pegged at 19%. They now have very different results so a short discussion around this might be of interest. Similarly the Ontario data had much lower PTB and SGA rates as they comment on. Could they also comment on what affect this might have on results (i.e. presumably given the high prevelance of GA around 39 weeks it might make the Ontario data more precise just based on the distribution being tighter.

They state parental consent was given. Does this mean both parents?

Fine for there to be an independent publication on the incidental findings but they might provide the numbers of positives found and which ones here.

What percent of infants had their heel sample collected prior to 24 hrs and did they see any differences in the profiles from these infants?

Is there a reference for there being little/no effect of storing the samples at ambient temperature (and ? humidity or are desiccants used?)

Line 133 should be T-cell not t-cell.

Could table 1 also report the number of LGA infants.

Overall this is nice report from an important study showing the possible use case for metabolic screening of gestational age in LMIC and reinforcing the known challenges in getting an appropriately timed US. It will inform future studies of this type and create a platform on which the true incidences of PTB and SGA can be reported and monitored with far greater accuracy.

Reviewer #2: This is an interesting study where the authors have tried to validate a metabolite prediction algorithm developed using data collected from a western population for estimating gestational age in a cohort at Siaya County, Kenya. The authors have used the refined version of the prediction algorithm. They have further analysed the validated models based on weeks that ultrasonography has been performed. Their model at collected a unique dataset using cutting edge methodology. The paper is generally well written and structured. However, in my opinion the paper has some limitations with regards to sample selection and reporting of results. Below I have provided few recommendations that could be improve the manuscript.

Comment 1:

To predict GA using metabolites and clinical covariates, authors have used heel prick (n=1039) and cord blood (n=1012) from the Kenya cohort. Are these samples paired? If yes, then how does the analysis behave in terms of paired samples? Is heel prick giving better RMSE than paired cord blood or vice- versa?

Comment 2:

To report GA prediction using metabolites at delivery, authors have used RMSE and MAE in model-1 and 2 among heel prick and cord blood samples. So far, RMSE is a widely used variable to report GA estimation in weeks. Here the predicted GA in weeks is reported through MAE in the result section. What are the precise reasons/thoughts?

Comment 3: Have the authors observed any under or overestimation of GA or preterm rate when original prediction algorithm is tested on the Kenya cohort samples?

6. PLOS authors have the option to publish the peer review history of their article (what does this mean?). If published, this will include your full peer review and any attached files.

**Do you want your identity to be public for this peer review?** For information about this choice, including consent withdrawal, please see our Privacy Policy.

Reviewer #1: **Yes: **Jeffrey C Murray

Reviewer #2: **Yes: **Pallavi Kshetrapal

---

## [Decision Letter · Decision Letter 1]

26 Oct 2022

Real world external validation of metabolic gestational age assessment in Kenya

PGPH-D-22-00803R1

Dear Dr. Wilson,

We are pleased to inform you that your manuscript 'Real world external validation of metabolic gestational age assessment in Kenya' has been provisionally accepted for publication in PLOS Global Public Health.

Best regards,

Ramachandran Thiruvengadam, M.D.,

Academic Editor

Reviewer Comments (if any, and for reference):

Reviewer's Responses to Questions

**Comments to the Author**

1. If the authors have adequately addressed your comments raised in a previous round of review and you feel that this manuscript is now acceptable for publication, you may indicate that here to bypass the “Comments to the Author” section, enter your conflict of interest statement in the “Confidential to Editor” section, and submit your "Accept" recommendation.

Reviewer #1: All comments have been addressed

Reviewer #2: All comments have been addressed

2. Does this manuscript meet PLOS Global Public Health’s publication criteria? Is the manuscript technically sound, and do the data support the conclusions? The manuscript must describe methodologically and ethically rigorous research with conclusions that are appropriately drawn based on the data presented.

Reviewer #1: Yes

Reviewer #2: Yes

3. Has the statistical analysis been performed appropriately and rigorously?

Reviewer #1: Yes

Reviewer #2: Yes

4. Have the authors made all data underlying the findings in their manuscript fully available (please refer to the Data Availability Statement at the start of the manuscript PDF file)?

Reviewer #1: Yes

Reviewer #2: Yes

5. Is the manuscript presented in an intelligible fashion and written in standard English?

Reviewer #1: Yes

Reviewer #2: Yes

6. Review Comments to the Author

Reviewer #1: Continues to be very nice work and they have addressed all the concerns expressed in the initial reviews.

Reviewer #2: All comments have been answered by the authors.

7. PLOS authors have the option to publish the peer review history of their article (what does this mean?). If published, this will include your full peer review and any attached files.

**Do you want your identity to be public for this peer review?** For information about this choice, including consent withdrawal, please see our Privacy Policy.

Reviewer #1: **Yes: **Jeffrey C Murray

Reviewer #2: No
